# Prevalence and factors associated with comorbidities in Iranian patients with type 2 diabetes: A national study

Koushan Hajiuni[1,2‡], Sarmad Salehi[1,3‡], Zahra Rajabli[4], Sina Azadnajafabad[1],
Samaneh Akbarpour[1,5], Samaneh Asgari[6‡*], Nazila Rezaei[1‡*]

**1** Non-Communicable Diseases Research Center, Endocrinology and Metabolism Population Sciences Institute, Tehran University of Medical Sciences, Tehran, Iran, **2** School of Medicine, Tehran University of Medical Sciences, Tehran, Iran, **3** Department of Epidemiology & Biostatistics, School of Public Health, Tehran University of Medical Sciences, Tehran, Iran, **4** Endocrinology and Metabolism Research Center, Endocrinology and Metabolism Clinical Sciences Institute, Tehran University of Medical Sciences, Tehran, Iran, **5** Sleep Breathing Disorders Research Center, Tehran University of Medical Sciences, Tehran, Iran, **6** Prevention of Metabolic Disorders Research Center, Research Institute for Metabolic and Obesity Disorders, Research Institute for Endocrine Sciences, Shahid Beheshti University of Medical Sciences, Tehran, Iran

‡ KH and SS are co-first authors on this work. KH, SS, SA and NR contributed equally to this work.
* nazila_r@yahoo.com (NR); asgari.samaneh@gmail.com (SA)

## Abstract

### Introduction

Type 2 diabetes (T2DM) is often accompanied by comorbidities such as hypertension (HTN), cardiovascular disease (CVD), and chronic kidney disease (CKD), which increase the disease burden and complicate its management. In Iran, where diabetes prevalence is growing, understanding the extent and determinants of these comorbidities is crucial for improving clinical care and informing national public health strategies.

### Materials and methods

We used data from the WHO STEPwise approach to non-communicable diseases risk factor surveillance (STEPS) 2021 to assess the prevalence of comorbidities among Iranian patients with T2DM. Comorbidity was defined as ≥ 2 of: HTN, CKD, history of CVD, or cancer. Multivariable logistic regression was done to identify the potential socio-demographic factors associated with comorbidities.

### Results

Of a total of 2,900 participants aged 25-70 years with T2DM (56.24% women), 27.00% (95% confidence interval (CI): 24.74–29.40) had no comorbidity, 39.82% (95% CI: 37.26–42.43) had one comorbidity (HTN: 79.62%, CKD: 10.66%, CVD: 8.64%, cancer: 1.09%), and 33.18% (95% CI: 30.58–35.89) had ≥ 2 comorbidities.

**Data availability statement:** The data supporting this study are not publicly available due to ethical restrictions and the presence of potentially identifiable patient information, as mandated by the Ethics Committee of the Endocrinology and Metabolism Research Institute (EMRI), Tehran University of Medical Sciences. Data may, however, be made available upon reasonable request to the Data Access Committee at the National Institute for Health Research (NIHR), Tehran University of Medical Sciences (nihr@tums.ac.ir). Requests must include a detailed research proposal and will be evaluated by the NIHR Data Access Committee in line with institutional and ethical guidelines; approved access may require a formal Data Use Agreement.

**Funding:** The author(s) received no specific funding for this work.

**Competing interests:** The authors have declared that no competing interests exist.

**Abbreviations:** STEPS, STEPwise approach to NCD risk factor surveillance; NCD, Non-communicable disease; CKD, Chronic kidney disease; CVD, Cardiovascular disease; WI, Wealth Index; eGFR, Estimated glomerular filtration rate; GPAQ, Global Physical Activity Questionnaire; ANOVA, One-way analysis of variance; HTN, Hypertension; FPG, Fasting plasma glucose; HbA1c, Glycated hemoglobin A1c; BMI, Body Mass Index; WC, Waist Circumference; NCDRC, Non-Communicable Diseases Research Center; EMRI, Endocrinology and Metabolism Research Institute; VIFs, Variance Inflation Factors.

The prevalence of comorbidity was significantly associated with male gender, age ≥ 60 years, living in a rural area, body mass index >30 kg/m2 (all p-values < 0.05). However, higher years of schooling, being employed, and physical activity ≥ 150 min/week were associated with lower odds of comorbidities.

## Discussion

Over 70% of Iranian adults have additional health conditions alongside diabetes, which significantly impact public health and underscore the need for personalized and multi-faceted preventive approaches.

## Introduction

Diabetes is a significant non-communicable disease (NCD) that contributes substantially to premature deaths associated with NCDs. Individuals with diabetes face an elevated risk of developing severe complications, leading to increased healthcare costs, diminished quality of life, and increased mortality rates [1]. Currently, 537 million adults aged 20–79 are living with diabetes, a figure that is expected to rise to 643 million by 2030 and 783 million by 2045. Over three-quarters of these individuals reside in low- and middle-income countries [2,3]. Diabetes is a major public health concern in Iran, with its prevalence among adults aged over 25 years showing a notable increase from 10.85% to 14.2% since 2016. Despite a high level of awareness and treatment, effective glycemic control remains a challenge. Factors such as age, obesity, urbanization, and socioeconomic status have been identified as significant contributors to both the prevalence and control of diabetes in the country [4].

Type 2 diabetes (T2DM) is frequently accompanied by additional chronic conditions that complicate management and increase healthcare burden [5]. Evidence suggests that nearly 70% of adults with diabetes have at least one comorbidity, and approximately 40% of older adults live with four or more [6,7]. Hypertension (HTN), chronic kidney disease (CKD), cardiovascular disease (CVD), and cancer are among the most common and clinically relevant comorbidities, each contributing to poorer health outcomes and higher healthcare utilization [8].

Comorbidities significantly affect glycemic control in patients with diabetes, often complicating management strategies. Studies indicate a strong correlation between high rates of comorbidity and poor glycemic outcomes, such as elevated glycated hemoglobin A1c (HbA1c) levels and a greater risk of severe hypoglycemia. Despite this correlation, some studies have reported no significant relationship between the number of comorbidities and the HbA1c level. HbA1c is a key measure for establishing glycemic targets and assessing the effectiveness of diabetes management. Reducing HbA1c levels can lead to clinical benefits, including fewer microvascular and cardiovascular complications [5].

Emerging evidence suggests that the number of comorbidities and specific comorbidity clusters may be more informative for clinical decision-making than examining

conditions individually. Clusters such as HTN–CKD or HTN–CVD have distinct implications for disease progression and treatment intensity.

Despite the growing recognition of multimorbidity as a key dimension of diabetes care, national-level evidence from Iran is limited. Most prior studies have relied on regional samples or focused on single comorbidities, and few have evaluated multimorbidity patterns or their relationship with diabetes awareness, treatment, and glycemic control. Consequently, the prevalence, distribution, and clinical impact of comorbidities among Iranian adults with diabetes remain insufficiently understood. Given these gaps, a comprehensive national assessment is needed to quantify the prevalence of comorbidities and multimorbidity patterns among Iranian adults with diabetes and to examine how these conditions influence diabetes management and glycemic outcomes. Addressing this evidence gap can support more effective resource allocation, enhance clinical decision-making, and guide targeted public health strategies aimed at improving diabetes care in Iran.

## Materials and methods

### Study design

This study used data from the 2021 Iran STEPS survey, a nationally representative cross-sectional study conducted in line with the World Health Organization's (WHO) STEPwise approach to monitoring NCD risk factors. The overall response rate of the 2021 Iran STEPS survey was 97.73%. The STEPS framework provides standardized guidance for monitoring key behavioral and biological risk factors across countries, thereby enabling comparability and trend analysis. The 2021 survey was implemented at the national level in Iran, ensuring the generalizability of the findings to the adult population of the country. Detailed information regarding the design and implementation of the 2021 Iran STEPS survey has been published elsewhere [6]. The survey consisted of two main phases—design and implementation, and data collection across three steps: questionnaire-based interviews, physical measurements, and laboratory assessments. Step 1 utilized the most recent WHO STEPS instrument (version 3.2) (27,874 individuals aged ≥18 completing the first step) [7] to collect behavioral and demographic data. Step 2 involved standardized measurements of weight, height, hip and waist circumferences, pulse rate, and blood pressure, as outlined in the survey protocol (27,745 individuals aged ≥ 18 completing the second step). In Step 3, laboratory analyses were conducted at the central survey facility via a validated autoanalyzer (Roche-Hitachi Cobas C311, High Technologies Corporation, Tokyo, Japan), which was approved by the national reference laboratory (18,119 individuals aged ≥ 25 completing the third step) [6].

### Study population

Of a total of 18,119 participants aged ≥ 25 years who completed the third step, 3,195 individuals with T2DM were included for the current analysis. After excluding subjects with missing data including age, sex, area, wealth index, job status, marital status, education levels, physical activity, smoking status, ever alcohol use, nutrition, weight, height, waist circumference (WC), wealth index, systolic/diastolic blood pressure (SBP/DBP) and serum creatinine (Cr) (n = 295), a total of 2,900 participants (1225 men and 1675 women) were considered for the current study analyses. Given that the missing data for covariates were < 10% (S1 Fig: maximum 5.1%), complete case analysis was employed, as the imputation may offer little advantage [8,9].

### Clinical and laboratory measurements

Demographic data regarding sex, age, education levels, occupation, marital status, basic health insurance, along with information on cigarette smoking, physical activity, alcohol use, medication use and medical history, were collected from participants through a standardized questionnaire derived from the STEPS survey. A positive history of CVD was defined by a "yes" answer to the following question: Have you ever had a heart attack or stroke? History of having cancer was defined as an affirmative answer ("Yes") to the question: In the past 12 months, has a doctor told you that you have cancer?

Participants' physical activity was evaluated utilizing the Global Physical Activity Questionnaire (GPAQ) [10]. The questionnaire includes 16 questions that measure the intensity, duration, and frequency of physical activity. The data collected from the GPAQ were then reported as metabolic equivalents (METs)–minutes per week [11]. Weight was recorded in kilograms while wearing minimal clothing utilizing a standard digital scale (Inofit), which was calibrated with a 5 kg reference scale before each use and whenever the device was relocated. Height was measured while standing barefoot using a measuring tape, with a possible error margin of 0.5 cm. WC was taken at the midpoint between the lowest rib and the hip, just above the umbilicus, along a straight line. SBPs and DBPs were recorded three times on the brachial artery, with a three-minute gap between each reading, after 15 minutes of rest in a seated position, using standard Beurer sphygmomanometers. The ultimate blood pressure value was finally calculated as the average of the second and third measures. Fasting blood glucose (FBS) was recorded utilizing commercial kits and the biochemistry auto analyzer (Cobas C311 Hitachi High–Technologies Corporation. Tokyo, Japan) and hemoglobin A1C (HbA1C) was recorded via High-Performance Liquid Chromatography (HPLC), approved by Health Reference. Inter- and intra-assay coefficients of variation were less than 3% for all the biomarkers used in this study [12]. Serum creatinine level was measured using the auto-analyzer (Roche-Hitachi Cobas C311, High–Technologies Corporation, Tokyo, Japan), an instrument approved by the reference laboratory [6].

## Definitions

Years of schooling were assessed by the years of formal education completed and categorized as 0, 1–6, 7–11, or ≥ 12 years. The wealth index (WI), used to assess socioeconomic status in this survey, was derived from household asset data collected through questionnaires. The index was then divided into five quintiles, ranging from the poorest (first quintile) to the wealthiest (fifth quintile) [6]. Smoking status was assessed via adapted WHO STEPS questions and defined as the current daily use of any tobacco product, including cigarettes, hookahs, pipes, smokeless tobacco, or electronic cigarettes [13]. Alcohol use was determined by asking participants whether they had consumed any alcoholic beverages in the past 12 months [14]. Physical activity was evaluated via the Global Physical Activity Questionnaire (GPAQ) version 2, developed by the WHO [15], and was categorized as low (< 150 minutes per week) or high (≥ 150 minutes per week) on the basis of total minutes of moderate to vigorous activity.

HTN was defined as a SBP ≥ 140 mmHg, a DBP ≥ 90 mmHg, or current use of antihypertensive medications [16]. Diabetes was defined according to the ADA guideline as FBS ≥ 126 mg/dL, HbA1C ≥ 6.5%, or the use of glucose-lowering medications, including oral medications or insulin [17]. Among individuals with diabetes, awareness was defined as having received a prior diagnosis by a healthcare provider, treatment was defined as current use of medication, and control was defined as achieving HbA1C < 7.0% [18]. Estimated glomerular filtration rate (eGFR) was calculated using the CKD-EPI creatinine equation developed by the CKD Epidemiology Collaboration [19], and CKD was defined as an eGFR < 60 mL/min/1.73 m² [20]. Diet score was defined based on thirteen nutritional indicators extracted from the STEPS questionnaire. These indicators reflected participants' usual eating patterns, including the number of meals and snacks consumed per day, weekly breakfast habit, daily portions of fruits, vegetables, and dairy products (along with dairy types), as well as consumption of red and processed meats, fish, whole grains, nuts, and sugar-sweetened drinks. Each component was independently evaluated by two nutritionists, and all item scores were normalized to a 0–1 scale.

Comorbidity was defined as the coexistence of ≥ 2 of the additional health conditions including HTN, CKD, history of CVD, or cancer alongside diabetes.

## Statistical analysis

Survey weights were applied to account for the complex sampling design, nonresponse, and poststratification. Specifically, weights were adjusted for non-response within age groups—an important consideration during the COVID-19 pandemic—along with missing data at each stage of the STEPS process. Poststratification weighting ensured that the

sample represented the Iranian population by age, sex, and residence location in 2016. The design effect and cluster-level variance were incorporated into the survey weights used in this study, and detailed information on the weighting procedures is available in the STEPS survey protocol [6]. To summarize the dietary information, all nutrition-related variables were entered into a principal component framework to identify the underlying patterns of eating behavior. The extracted components were combined to generate a single composite diet score, with each component contributing proportionally according to its explained variance. For subsequent analyses, the final continuous diet score was divided into tertiles, creating three groups that reflected low, moderate, and high dietary quality across the study population.

Descriptive statistics, including weighted prevalence estimates and 95% confidence intervals, were calculated. The results for continuous variables are reported as the means, and categorical variables are expressed as proportions. Categorical variables were compared via chi-square tests, whereas continuous variables were compared via one-way analysis of variance (ANOVA).

Multinomial stepwise logistic regression models were used to examine the factors associated with the number of comorbid conditions among individuals with diabetes. Candidate variables for inclusion in the model were age, sex, residential area, years of schooling, BMI, marital status, employment status, physical activity, wealth index, smoking, alcohol consumption, basic health insurance, presence of a glucometer at home, and diet score. Multi-collinearity among predictors was assessed using variance inflation factors (VIFs), and all VIFs were below 2, indicating no significant multi-collinearity. Variables were initially selected through a stepwise approach with a significance threshold of p-value < 0.2. Subsequently, significant variables were entered into the multivariable model, and those with a p-value greater than 0.05 were excluded. The final multivariable model retained age, sex, BMI, physical activity, residential area, employment status, and years of schooling as independent variables. Potential interactions between age/sex and other covariates were tested by including relevant multiplicative terms in the final regression model. Notably, no significant gender and age interactions were observed (the minimum interaction p-value of 0.01 exceeded the Bonferroni-corrected significance level of 0.005). However, all analyses were conducted separately for men, women, and the total population. All the statistical analyses were performed via Stata version 14.2 (StataCorp LLC, College Station, TX, USA) [21], and the figures were produced via Microsoft Excel 2016 [22].

### Ethics approval and consent to participate

All participants were fully informed about the study's objectives and procedures. Participation was voluntary, and written informed consent was obtained from each individual. To ensure confidentiality, the final dataset was de-identified prior to analysis, with access restricted to the principal investigator and the data manager. The study was approved by the Research Ethics Committee of the Endocrinology and Metabolism Research Institute (EMRI), Tehran University of Medical Sciences (Approval ID: IR.TUMS.EMRI.REC.1403.102), and conducted in accordance with the Declaration of Helsinki. COVID-19 infection prevention protocols were strictly followed during data collection.

### Results

A total of 2,900 participants (women: 56.24%), mean age of 57.96 years (95% CI: 57.28–58.64), were included in the current data analysis. At the overall population level, HTN was the most prevalent comorbidity (63.46%, 95% CI: 60.89–65.96), followed by CKD (27.44%, 95% CI: 25.04–29.97), CVD (20.61%, 95% CI: 18.48–22.92), and cancer (1.88%, 95% CI: 1.41–2.51). The overall prevalence of multi-morbidity (≥2 comorbidities) among adults with T2DM was 33.18% (95% CI: 30.58–35.89). Among individuals with a single comorbidity, HTN was the most common condition (79.62%, 95% CI: 76.37–82.52), followed by CKD (10.66%, 95% CI: 8.64–13.09), CVD (8.64%, 95% CI: 6.62–11.19), and cancer (1.09%, 95% CI: 0.58–2.02). HTN also clustered most frequently with other conditions. The most common dyad was HTN and CKD (57.67%, 95% CI: 51.60–63.52), followed by HTN and CVD (35.66%, 95% CI: 30.06–41.68), CVD and CKD (4.72%, 95% CI: 2.76–7.96), HTN and cancer (1.34%, 95% CI: 0.72–2.48), CKD and cancer (0.54%, 95% CI: 0.19–1.51), and

CVD and cancer (0.07%, 95% CI: 0.02–0.29). The combination of HTN, CKD, and CVD was the most prevalent triad among those with three or more comorbidities (88.95%, 95% CI: 82.94–93.02) (Table 1).

The crude mean HbA1c values for individuals with varying numbers of health conditions increased from 7.76 (95% CI: 7.60–7.93) for those with no additional conditions to 8.09 (95% CI: 7.80–8.38) for those with three or more conditions. After adjusting for anti-diabetic medications, the mean HbA1C values showed a slight decrease across the categories, from 7.85 for no additional conditions to 7.95 (95% CI: 7.65–8.24) for those with ≥ 3 conditions (Fig 1A). Similarly, for FBS, the crude mean levels range from 149.99 (95% CI: 144.63–155.36) mg/dL in individuals with no additional conditions to 151.53 (95% CI: 142.19–160.88) mg/dL in those with three or more conditions, with anti-diabetic adjustments yielding slightly lower values across the categories (Fig 1B).

## Hemoglobin A1C (HbA1c); Fasting blood sugar (FBS)

To characterize patterns of diabetes status, prevalence of patients with awareness, on treatment, and those with glycemic control across different levels of comorbidities, weighted prevalence estimates with 95% confidence intervals were calculated (Table 2). Among individuals with diabetes, the weighted prevalence of undiagnosed diabetes was 12.58% (95% CI: 7.72–19.83) in those with zero morbidities, 7.50% (95% CI: 5.36–10.41) in those with one morbidity, 8.91% (95% CI: 6.00–13.04) in individuals with two morbidities, and 3.59% (95% CI: 1.36–9.11) among those with three or more morbidities. The prevalence of being untreated was 4.71% (95% CI: 2.71–8.06), 4.98% (95% CI: 3.45–7.15), 6.48% (95% CI: 3.72–11.06), and 6.76% (95% CI: 2.68–16.02) among participants with zero, one, two and ≥ 3 morbidities, respectively. Among individuals receiving treatment, the proportion of uncontrolled diabetes was 59.05% (95% CI: 51.59–66.12) in those with no morbidities, 62.49% (95% CI: 57.50–67.24) in those with one morbidity, 63.19% (95% CI: 56.73–69.21) in

**Table 1. Distribution of comorbidities among Iranian adults with type 2 diabetes (N = 2900): STEPS 2021.**

| Variable | Components | N | Prevalence (95% CI) |
|---|---|---|---|
| 0 comorbidity (N = 766) | T2DM | 766 | 100.0 |
| 1 comorbidity (N = 1210) | HTN | 952 | 79.62 (76.37, 82.52) |
| | CKD | 145 | 10.66 (8.64, 13.09) |
| | CVD | 100 | 8.64 (6.62, 11.19) |
| | Cancer | 13 | 1.09 (0.58, 2.02) |
| 2 comorbidities (N = 712) | HTN + CKD | 437 | 57.67 (51.60, 63.52) |
| | HTN + CVD | 229 | 35.66 (30.06, 41.68) |
| | HTN +Cancer | 12 | 1.34 (0.72, 2.48) |
| | CKD + CVD | 27 | 4.72 (2.76, 7.96) |
| | CKD +Cancer | 5 | 0.54 (0.19, 1.51) |
| | CVD +Cancer | 2 | 0.07 (0.02, 0.29) |
| 3 comorbidities (N = 206) | HTN + CKD + CVD | 183 | 88.95 (82.94, 93.02) |
| | HTN + CKD+ Cancer | 13 | 6.39 (3.50, 11.40) |
| | HTN + CVD+ Cancer | 9 | 4.32 (1.99, 9.13) |
| | CVD + CKD+ cancer | 1 | 0.34 (0.05, 2.41) |
| 4 comorbidities (N = 6) | HTN + CKD + CVD+ Cancer | 6 | 0.19 (0.01,0.45) |
| Multi-morbidity | ≥2 co-morbidities | 924 | 33.18 (30.58, 35.89) |

HTN: hypertension; T2DM: type 2 diabetes; CKD: chronic kidney disease; CVD: cardiovascular disease; N: number.

A

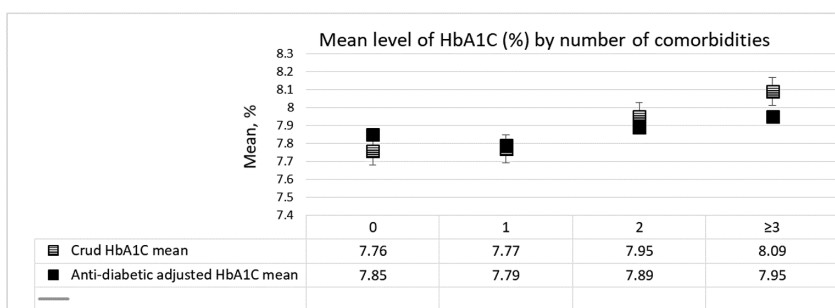

B

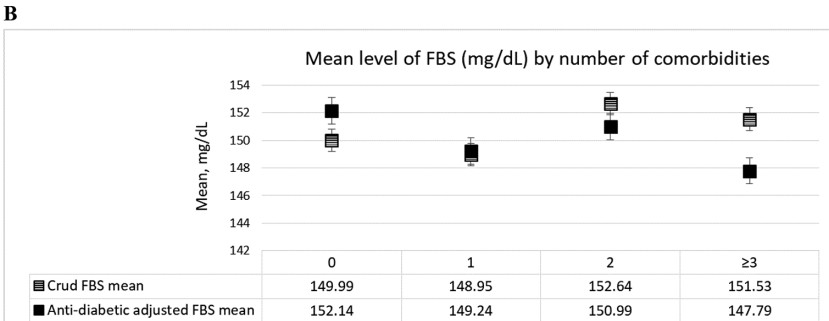

**Fig 1. Mean level of HbA1c and FPG by number of comorbidities in the diabetic population (n = 2900).** A: Crude and anti-diabetic adjusted mean values of HbA1c (%). B: Crude and anti-diabetic adjusted mean values of FBS (mg/dL).

**Table 2. Diabetes status according to the number of morbidities.**

| Diabetes status | | | Number of morbidities | | | |
|---|---|---|---|---|---|---|
| | | | 0 (n = 766) | 1 (n = 1210) | 2 (n = 712) | ≥3 (n = 212) |
| Unaware (undiagnosed) (N = 1387) | | | 12.58 (7.72, 19.83) | 7.50 (5.36, 10.41) | 8.91 (6.00, 13.04) | 3.59 (1.36, 9.11) |
| Aware (N = 1513) | No treatment(N = 88) | | 4.71 (2.71, 8.06) | 4.98 (3.45, 7.15) | 6.48 (3.72, 11.06) | 6.76 (2.68, 16.02) |
| | under treatment (N = 1425) | Uncontrolled (N = 1014) | 59.05 (51.59, 66.12) | 62.49 (57.50, 67.24) | 63.19 (56.73, 69.21) | 65.98 (55.63, 75.00) |
| | | Controlled (N = 411) | 23.67 (18.60, 29.62) | 25.02 (20.91, 29.64) | 21.42 (16.57, 27.24) | 23.67 (15.97, 33.61) |

Awareness: Among individuals with diabetes was defined as having received a prior diagnosis by a healthcare provider.

Under treatment: was defined as current use of medication.

Control: was defined as achieving HbA1c < 7.0%.

Column weighted prevalence (95% CI: confidence interval) was reported.

those with two morbidities, and 65.98% (95% CI: 55.63–75.00) among participants with three or more morbidities. The corresponding prevalence of controlled diabetes was 23.67% (95% CI: 18.60–29.62), 25.02% (95% CI: 20.91–29.64), 21.42% (95% CI: 16.57–27.24), and 23.67% (95% CI: 15.97–33.61) across the same morbidity categories, respectively.

The distribution of key clinical and metabolic indicators across levels of comorbidity is presented in Table 3. Among Iranian adults with diabetes, 27.0% had no comorbidities, 39.82% had one, and 33.18% had two or more. Mean SBP increased progressively with the number of comorbidities, rising from 120.46 mmHg (95% CI: 119.33–121.58) among

**Table 3. Prevalence of comorbidity according to health-related variables on Iranian patients with diabetes (N = 2900): Steps 2021.**

| variables | N | Total population (N = 2900) | | Number of Comorbidities along with Diabetes | | | | | | P value |
|---|---|---|---|---|---|---|---|---|---|---|
| | | | | 0 (n = 766) | | 1 (n = 1210) | | ≥2 (n = 924) | | |
| Total % | 2900 | 100.0 | | 27.00 | 24.74, 29.40 | 39.82 | 37.26, 42.43 | 33.18 | 30.58, 35.89 | |
| | | **Mean** | **95% CI** | **Mean** | **95% CI** | **Mean** | **95% CI** | **Mean** | **95% CI** | |
| Age, years | 2900 | 58.41 | 57.75, 59.08 | 51.70 | 50.67, 52.73 | 57.86 | 56.92, 58.80 | 64.54 | 63.50, 65.57 | <0.001 |
| BMI (kg/m²) | 2900 | 29.29 | 29.03, 29.55 | 28.47 | 28.04, 28.90 | 29.58 | 29.23, 29.94 | 29.60 | 29.06, 30.14 | <0.001 |
| SBP, mmHg | 2900 | 136.27 | 135.29, 137.26 | 120.46 | 119.33, 121.58 | 140.22 | 138.74, 141.70 | 144.40 | 142.74, 146.06 | <0.001 |
| DBP, mmHg | 2900 | 81.03 | 80.48, 81.59 | 74.88 | 74.07, 75.69 | 84.11 | 83.24, 84.99 | 82.35 | 81.41, 83.29 | <0.001 |
| FBS, mg/dL | 2900 | 150.38 | 147.64, 153.12 | 149.99 | 144.63, 155.36 | 148.95 | 144.75, 153.15 | 152.41 | 147.49, 157.33 | 0.004 |
| | | **%** | **95% CI** | **%** | **95% CI** | **%** | **95% CI** | **%** | **95% CI** | |
| HbA1c, | 2900 | 7.84 | 7.75, 7.93 | 7.76 | 7.60, 7.93 | 7.77 | 7.64, 7.91 | 7.98 | 7.82, 8.15 | <0.001 |
| Age (years) | | | | | | | | | | <0.001 |
| <60 | 1531 | 52.43 | 49.72, 55.12 | 39.16 | 35.78, 42.65 | 41.81 | 38.34, 45.36 | 19.04 | 16.39, 22.00 | |
| ≥60 | 1369 | 47.57 | 44.88, 50.28 | 13.61 | 11.06, 16.63 | 37.62 | 33.94, 41.46 | 48.77 | 44.67, 52.88 | |
| Sex | | | | | | | | | | 0.330 |
| Female | 1675 | 56.63 | 53.96, 59.27 | 26.25 | 23.34, 29.38 | 41.57 | 38.11, 45.11 | 32.18 | 28.67, 35.92 | |
| Male | 1225 | 43.37 | 40.73, 46.04 | 27.99 | 24.50, 31.77 | 37.53 | 33.79, 41.43 | 34.48 | 30.69, 38.47 | |
| Residential area | | | | | | | | | | 0.421 |
| Urban | 2116 | 79.53 | 77.75, 81.19 | 26.40 | 23.74, 29.24 | 40.37 | 37.32, 43.49 | 33.23 | 30.13, 36.48 | |
| Rural | 784 | 20.47 | 18.81, 22.25 | 29.35 | 25.65, 33.35 | 37.68 | 33.78, 41.75 | 32.97 | 29.08, 37.11 | |
| Years of schooling (years) | | | | | | | | | | <0.001 |
| <6 | 813 | 22.76 | 20.92, 24.70 | 13.60 | 11.00, 16.69 | 40.53 | 36.40, 44.79 | 45.87 | 41.62, 50.19 | |
| 1_6 | 1004 | 33.63 | 31.11, 36.24 | 24.52 | 21.12, 28.27 | 42.58 | 38.13, 47.16 | 32.90 | 28.06, 38.13 | |
| 7_11 | 409 | 15.48 | 13.63, 17.53 | 35.18 | 29.28, 41.57 | 37.07 | 31.07, 43.51 | 27.75 | 21.11, 35.55 | |
| ≥12 | 674 | 28.13 | 25.58, 30.84 | 36.33 | 30.96, 42.06 | 37.45 | 31.96, 43.27 | 26.23 | 21.66, 31.37 | |
| BMI (kg/m²) | | | | | | | | | | 0.020 |
| <25 | 554 | 20.18 | 18.04, 22.51 | 32.34 | 27.11, 38.05 | 34.96 | 29.51, 40.83 | 32.70 | 26.55, 39.52 | |
| 25-30 | 1137 | 37.54 | 35.06, 40.10 | 29.20 | 25.50, 33.21 | 39.95 | 36.13, 43.91 | 30.84 | 27.38, 34.54 | |
| >30 | 1209 | 42.28 | 39.61, 44.99 | 22.50 | 19.31, 26.05 | 42.02 | 37.85, 46.29 | 35.48 | 31.22, 39.98 | |
| Waist circumference (cm) | | | | | | | | | | <0.001 |
| <95 | 760 | 27.92 | 25.46, 30.52 | 37.77 | 32.57, 43.27 | 38.14 | 33.06, 43.50 | 24.09 | 19.61, 29.22 | |
| ≥95 | 2140 | 72.08 | 69.48, 74.54 | 22.83 | 20.55, 25.29 | 40.47 | 37.55, 43.46 | 36.70 | 33.64, 39.87 | |
| Marital status | | | | | | | | | | <0.001 |
| Single or widow | 507 | 17.22 | 15.36, 19.27 | 20.20 | 15.70, 25.60 | 36.58 | 30.97, 42.59 | 43.22 | 37.18, 49.47 | |
| Married | 2393 | 82.78 | 80.73, 84.64 | 28.42 | 25.88, 31.11 | 40.49 | 37.65, 43.40 | 31.09 | 28.23, 34.10 | |
| Employment status | | | | | | | | | | <0.001 |
| Unemployed & retired & unpaid job | 2194 | 75.83 | 73.56, 77.97 | 22.69 | 20.24, 25.34 | 40.47 | 37.49, 43.52 | 36.84 | 33.73, 40.07 | |
| Employed | 706 | 24.17 | 22.03, 26.44 | 40.53 | 35.60, 45.66 | 37.78 | 32.93, 42.90 | 21.69 | 17.82, 26.12 | |
| Physical activity (min/week) | | | | | | | | | | <0.001 |
| <150 | 1710 | 60.27 | 57.66, 62.83 | 24.09 | 21.35, 27.07 | 38.62 | 35.25, 42.11 | 37.29 | 33.76, 40.95 | |

*(Continued)*

Table 3. (Continued)

| variables | N | Total population (N = 2900) | | Number of Comorbidities along with Diabetes | | | | | | P value |
|---|---|---|---|---|---|---|---|---|---|---|
| | | | | 0 (n = 766) | | 1 (n = 1210) | | ≥2 (n = 924) | | |
| ≥ 150 | 1190 | 39.73 | 37.17, 42.34 | 31.42 | 27.69, 35.40 | 41.63 | 37.79, 45.59 | 26.95 | 23.38, 30.84 | |
| Wealth index % | | | | | | | | | | 0.022 |
| 1st quintile (lowest) | 663 | 20.22 | 18.06, 22.58 | 21.88 | 17.91, 26.45 | 37.75 | 32.33, 43.50 | 40.37 | 33.75, 47.35 | |
| 2nd quintile | 622 | 22.49 | 20.35, 24.79 | 25.74 | 21.20, 30.87 | 38.61 | 33.33, 44.18 | 35.65 | 30.47, 41.18 | |
| 3rd quintile | 583 | 17.85 | 16.17, 19.65 | 26.95 | 22.68, 31.70 | 42.19 | 37.32, 47.21 | 30.86 | 26.27, 35.86 | |
| 4th quintile | 569 | 19.66 | 17.68, 21.81 | 26.69 | 22.02, 31.94 | 43.87 | 38.16, 49.75 | 29.44 | 24.60, 34.80 | |
| 5th quintile (highest) | 463 | 19.77 | 17.48, 22.29 | 34.04 | 27.69, 41.02 | 37.13 | 30.77, 43.97 | 28.83 | 22.81, 35.70 | |
| Smoking stats | | | | | | | | | | 0.974 |
| no | 2545 | 87.52 | 85.66, 89.16 | 27.08 | 24.63, 29.67 | 39.84 | 37.10, 42.64 | 33.09 | 30.32, 35.97 | |
| yes | 355 | 12.48 | 10.84, 14.34 | 26.49 | 20.94, 32.90 | 39.68 | 32.82, 46.96 | 33.83 | 26.60, 41.90 | |
| Ever alcohol consumption | | | | | | | | | | 0.632 |
| no | 2846 | 98.17 | 97.51, 98.65 | 27.00 | 24.70, 29.42 | 39.94 | 37.35, 42.59 | 33.06 | 30.43, 35.81 | |
| yes | 54 | 1.83 | 1.35, 2.49 | 27.31 | 15.71, 43.09 | 33.31 | 21.12, 48.24 | 39.38 | 25.44, 55.29 | |
| Basic health insurance | | | | | | | | | | 0.235 |
| no | 168 | 6.36 | 5.07, 7.97 | 35.44 | 25.27, 47.12 | 37.54 | 27.00, 49.42 | 27.02 | 17.64, 39.02 | |
| yes | 2732 | 93.64 | 92.03, 94.93 | 26.43 | 24.13, 28.87 | 39.97 | 37.35, 42.65 | 33.60 | 30.92, 36.39 | |
| Glucometer at home | | | | | | | | | | 0.014 |
| no | 1775 | 59.30 | 56.66, 61.89 | 29.28 | 26.29, 32.46 | 40.39 | 37.02, 43.86 | 30.33 | 26.84, 34.06 | |
| yes | 1125 | 40.70 | 38.11, 43.34 | 23.69 | 20.31, 27.45 | 38.98 | 35.11, 42.98 | 37.33 | 33.48, 41.34 | |
| Diet score | | | | | | | | | | 0.292 |
| 1st tertile (lowest) | 968 | 31.01 | 28.55, 33.58 | 24.63 | 21.06, 28.59 | 38.93 | 34.43, 43.64 | 36.43 | 31.43, 41.75 | |
| 2nd tertile | 967 | 33.69 | 31.21, 36.26 | 26.17 | 22.48, 30.23 | 41.53 | 37.08, 46.12 | 32.30 | 28.14, 36.76 | |
| 3rd tertile (highest) | 965 | 35.30 | 32.78, 37.91 | 29.88 | 25.78, 34.33 | 38.96 | 34.78, 43.31 | 31.16 | 27.06, 35.57 | |

CI: confidence interval; min, minutes; BMI: Body mass index; SBP/DBP: systolic/diastolic blood pressure; FBS: fasting blood sugar; Hemoglobin A1C: HbA1c.

Data have been weighted to account for overall non-response, non-response at each step, and sample distribution across provinces, adjusted for age, sex, and area of residence. The p-value was calculated using the chi-square test.

For continues variables mean (95% CI) and for the categorical variables the row percentages (95% CI) was reported.

CI: confidence interval; min, minutes; BMI: Body mass index; Data have been weighted to account for overall non-response, non-response at each step, and sample distribution across provinces, adjusted for age, sex, and area of residence. The p-value was calculated using the chi-square test.

The reported percentages represent row-wise prevalence.

those with no comorbidity to 140.22 mmHg (95% CI: 138.74–141.70) in individuals with one comorbidity, and 144.40 mmHg (95% CI: 142.74–146.06) among those with two or more (p < 0.001). A similar pattern was observed for DBP, with mean DBP increasing from 74.88 mmHg (95% CI: 74.07–75.69) in the no-comorbidity group to 84.11 mmHg (95% CI: 83.24–84.99) in individuals with one, followed by a slight decrease to 82.35 mmHg (95% CI: 81.41–83.29) among those with ≥ 2 comorbidities (p < 0.001). FBS and HbA1c levels were also higher among participants with multiple comorbidities. Mean FBS increased from 149.99 mg/dL (95% CI: 144.63–155.36) in those without comorbidities to 152.41 mg/dL (95% CI: 147.49–157.33) among those with ≥ 2 (p = 0.004). Similarly, mean HbA1c rose from 7.76% (95% CI: 7.60–7.93) to 7.98% (95% CI: 7.82–8.15) across the same groups (p < 0.001).

Comorbidities were significantly associated with older age, lower educational level, higher BMI, increased waist circumference, unemployment or retirement, lower physical activity, and lower wealth index (all p < 0.05). Participants aged ≥ 60 years and those with < 6 years of schooling showed the highest prevalence of ≥ 2 comorbidities. In contrast,

sex, residential area, smoking, alcohol use, and basic health insurance showed no significant associations. Ownership of a home glucometer was associated with higher comorbidity (p = 0.014).

Table 4 presents the results of the stepwise multinomial logistic regression model to identify the association of socio-demographic and lifestyle determinants of comorbidities. Multivariable analysis revealed that participants aged 60 years or older had significantly greater odds of having one or two comorbidities compared to those under 60 (OR: 2.00, 95% CI: 1.42–2.81, and OR: 5.10, 95% CI: 3.50–7.45, respectively; p < 0.001). Compared to females, male participants had approximately twice the odds of having two comorbidities (OR: 1.87, 95% CI: 1.23–2.82, p < 0.001). Engaging in ≥ 150 minutes of physical activity per week resulted in lower odds of having two comorbidities by about 30% (OR: 0.72, 95% CI: 0.52–0.99; p = 0.045). Rural vs. urban residence was associated with higher odds of having one comorbid condition (OR: 1.41, 95% CI: 1.08–1.84; p = 0.013). Participants with higher education were less likely to have one or two comorbidities. Obesity increased the OR of having one and two comorbidities, compared to those with normal weight (OR: 1.93, 95% CI:

**Table 4. Stepwise Multinomial Logistic Regression Analysis of Factors Associated with Comorbidity in the Diabetic Population.**

| variables | 1 comorbidity vs. 0 comorbidity | | Interaction with age† | Interaction with sex† | 2 comorbidities vs. 0 comorbidity | | Interaction with age | Interaction with sex |
|---|---|---|---|---|---|---|---|---|
| | OR (95% CI) | P value | P value | P value | OR (95% CI) | P value | P value | P value |
| Age (years) | | | – | 0.35 | | | – | 0.67 |
| <60 | Reference | | | | Reference | | | |
| ≥60 | **2.00 (1.42, 2.81)** | **<0.001** | | | **5.10 (3.50, 7.45)** | **<0.001** | | |
| Sex | | | 0.35 | – | | | 0.67 | – |
| Female | Reference | | | | Reference | | | |
| Male | 1.35 (0.93, 1.96) | 0.114 | | | **1.87 (1.23, 2.82)** | **<0.001** | | |
| Physical activity (min/week) | | | 0.33 | 0.01 | | | 0.32 | 0.06 |
| <150 | Reference | | | | Reference | | | |
| ≥150 | 0.97 (0.74, 1.27) | 0.820 | | | **0.72 (0.52, 0.99)** | **0.045** | | |
| Residential area | | | 0.55 | 0.17 | | | 0.78 | 0.24 |
| Urban | Reference | | | | Reference | | | |
| Rural | **1.41 (1.08, 1.84)** | **0.013** | | | 1.26 (0.92, 1.72) | 0.144 | | |
| Years of schooling (years) | | | 0.65 | 0.26 | | | 0.56 | 0.36 |
| <6 | Reference | | | | Reference | | | |
| 1_6 | **0.64 (0.45, 0.91)** | **0.013** | | | **0.53 (0.36, 0.78)** | **0.001** | | |
| 7_12 | **0.45 (0.29, 0.69)** | **<0.001** | | | **0.46 (0.26, 0.79)** | **0.005** | | |
| >12 | **0.42 (0.27, 0.65)** | **<0.001** | | | **0.37 (0.23, 0.58)** | **<0.001** | | |
| BMI (kg/m²) | | | 0.34 | 0.54 | | | 0.76 | 0.86 |
| <25 | Reference | | | | Reference | | | |
| 25-30 | 1.37 (0.95, 1.97) | 0.093 | | | 1.29 (0.82, 2.01) | 0.271 | | |
| >30 | **1.93 (1.32, 2.83)** | **0.001** | | | **2.10 (1.32, 3.35)** | **0.002** | | |
| Employment status | | | 0.89 | 0.37 | | | 0.70 | 0.04 |
| Unemployed & retired & unpaid job | Reference | | | | Reference | | | |
| Employed | **0.61 (0.42, 0.91)** | **0.015** | | | **0.45 (0.28, 0.71)** | **0.001** | | |

OR, odds ratio; CI, confidence interval; BMI, body mass index.

Data have been weighted to account for overall non-response, non-response at each step, and sample distribution across provinces, adjusted for age, sex, and area of residence.

Stepwise selection was used for variable selection, with an entry p-value of 0.2 and a removal p-value of 0.05.

Statistically significant (P value<0.05) statistics are bold.

†The p-values for interaction terms (age × variable and sex × variable) were evaluated against a Bonferroni-corrected significance threshold of 0.005 (α = 0.05 adjusted for 11 tested models). None of the interaction terms remained statistically significant after this correction.

1.32–2.83 and OR: 2.10, 95% CI: 1.32–3.35, respectively). Employment was significantly associated with a lower risk of having one or two comorbidities (OR: 0.61, 95% CI: 0.42–0.91 and OR: 0.45, 95% CI: 0.28–0.71, respectively).

The results of the analysis of factors associated with comorbidity in individuals with diabetes, stratified by sex (male and female), are also presented in S1 and S2 Tables. The results indicated that among females with diabetes, residential area and employment status were not associated with comorbidity.

## Discussion

This study is one of the first nationally representative studies in Iran that assessed comorbidity patterns among patients with diabetes using data from the 2021 STEPS survey. It highlights important differences in diabetes awareness, treatment, and control across groups stratified by the number of comorbidities. In addition, this study identified older age (> 60 years), male sex, low physical activity, rural residence, lower education, obesity, and unemployment as factors associated with having comorbidities among patients with diabetes.

Our findings indicated that approximately 70% of individuals with diabetes had at least one comorbid condition, with HTN being the most prevalent, followed by CKD, CVD, and cancer. It should also be noted that CVD and cancer were based on self-reported diagnoses, whereas HTN, CKD, and diabetes were identified through clinical or laboratory criteria. This difference in disease ascertainment may potentially lead to an underestimation of CVD and cancer and influence the relative distribution of comorbidity types. Among clusters of comorbidities, the most frequent dyad was HTN and CKD, and the most common triad was HTN, CKD, and CVD. These patterns suggest shared pathophysiological pathways, such as systemic inflammation, endothelial dysfunction, and insulin resistance, highlighting the interrelated nature of cardio-renal complications in diabetes and their implications for morbidity and mortality [23,24].

Regarding diabetes status, participants without comorbidities presented a higher rate of unawareness than those with one or more comorbid conditions. These findings suggest that individuals with more comorbidities may experience health-related issues more frequently, leading to earlier detection of diabetes. Conversely, individuals without comorbidities might perceive themselves as healthier, possibly delaying medical checkups and diabetes diagnosis. It should be noted that some relevant factors were not measured in the STEPS survey, including medication adherence, access to healthcare services, treatment intensity, and genetic predisposition. These unmeasured factors may partly influence both comorbidity patterns and glycemic control, potentially leading to residual confounding when interpreting the observed associations.

Our findings confirm the high burden of multimorbidity in the diabetic population and mirror the global patterns reported in countries such as Japan and India, where the prevalence of comorbidities exceeds 80% [25,26]. Furthermore, the dominance of HTN among comorbidities aligns with international patterns, including 47.6% in the EU population [26] and 50.4% in a Bangladeshi cohort [27].

Age was a powerful determinant: participants aged ≥ 60 years had 5.1 times greater odds of having two or more comorbidities, which aligns with studies showing that aging increases vulnerability to multiple chronic conditions due to cumulative metabolic and vascular damage [28–30]. In a U.S. cohort, 88.5% of adults with diabetes aged ≥ 65 had two or more comorbidities, and Italian data indicated that older adults accounted for 80% of total comorbidity-related healthcare costs [30,31]. These findings underscore the importance of age-stratified screening and care models.

Male sex was associated with increased odds of multimorbidity, consistent with the findings of previous studies [28,29,32]. One possible explanation is the earlier age of diabetes diagnosis in men, leading to a longer duration of disease and more exposure to risk factors [29]. However, sex-specific patterns may vary by region and comorbidity type. For example, in Pakistan, women are more likely to have three or more comorbidities than men (aOR = 2.32) [33].

Socioeconomic and behavioral factors also play critical roles. Participants with lower education levels had a significantly greater prevalence of multiple comorbidities, possibly reflecting disparities in health knowledge and access to care. Unemployment was associated with a greater burden of comorbidities, which is in line with Bangladeshi data indicating

that unemployment tripled the odds of multimorbidity [34]. Physical activity of ≥ 150 minutes/week was associated with a 28% reduction in the odds of having two or more comorbidities, consistent with clinical trials that have shown exercise to reduce HbA1c by 0.7% and cardiovascular risk by 26% [31,35,36].

Waist circumference appeared to be a stronger predictor of comorbidities than BMI. Among patients with WC ≥ 95 cm, 36.7% had ≥ 2 comorbidities, and 40.47% had one comorbid condition. These findings support prior research indicating a 37% increase in diabetes risk per 3 cm increase in waist circumference [34,36,37]. They also highlight the value of central obesity markers in identifying high-risk individuals.

Marital status also affected the burden of comorbidities. The prevalence of having no comorbidities was significantly higher among married participants compared to singles (28.42% vs. 20.20%), which is consistent with studies linking marriage to improved chronic disease outcomes through better treatment adherence and social support [38,39]. However, divorced individuals remain a high-risk group for poor diabetes outcomes and mortality [39], indicating that relationship status may affect both the incidence and progression of comorbidities.

Our results also demonstrated that glycemic control worsens as the comorbidity burden increases. Individuals with two or more comorbidities had the highest mean FPG (152.41 mg/dL) and HbA1c (7.98%), whereas those without comorbidities had slightly lower values (FPG: 149.99 mg/dL; HbA1c: 7.76%). Although these differences are modest, they suggest a trend consistent with prior studies that have shown that comorbidities complicate diabetes management [40,41]. Notably, the group with ≥ 2 comorbidities also had the highest proportion of uncontrolled diabetes (63.83%).

In this study, individuals with a single comorbidity demonstrated slightly better glycemic control (25.02%) compared with those without any comorbidities (23.67%). Although this difference was small and not necessarily clinically meaningful, it may reflect differences in health-seeking behavior. Patients who already have one diagnosed chronic condition tend to have more frequent medical contact, more regular monitoring, and possibly higher adherence to treatment recommendations [42]. In contrast, individuals without comorbidities may perceive themselves as healthier and may have fewer interactions with the healthcare system. However, when the number of comorbidities increases, the complexity of treatment, potential medication interactions, and overall disease burden appear to outweigh these benefits, resulting in poorer glycemic control among those with multiple comorbid conditions [43,44]. These findings emphasize the importance of personalized management plans, especially for patients with multiple health conditions.

In our analysis, rural residence was associated with a higher odds of having *one* comorbid condition, although this association was not statistically significant for having two or more comorbidities. This finding aligns with studies that have shown regional differences in comorbidity factors. In Ethiopia, urban residents are more likely to be multimorbid, reflecting variations in urban lifestyle risk factors [33]. This pattern suggests that geographic differences may influence the early emergence of comorbid conditions but may play a less prominent role as the disease burden becomes more complex. Limited availability of preventive services and routine screening in rural areas may contribute to the earlier detection of single chronic conditions such as HTN or CKD. However, the development of multiple comorbidities likely reflects broader factors including age, obesity, and socioeconomic determinants that are not solely dependent on place of residence.

## Strengths and limitations

This study benefits from a large, nationally representative sample from the 2021 Iranian STEPS survey. The diverse demographic and geographic coverage support reliable statistical analysis and offers helpful information regarding the prevalence of multimorbidity among diabetic patients. Additionally, the study's focus on the co-occurrence of chronic conditions adds an important contribution to the growing body of literature on multimorbidity and its implications for future healthcare models.

Despite these strengths, several limitations should be acknowledged. First, the cross-sectional design restricts causal inference and limits the ability to assess temporal relationships or disease progression. Second, reliance on self-reported diagnoses and lifestyle behaviors may introduce recall or social desirability bias, potentially leading to misclassification or

underestimation of comorbidity prevalence. Third, although adjustments were made for several confounders, unmeasured factors such as genetic predisposition, duration of diabetes, detailed dietary intake, or environmental exposures may still influence the observed associations. Fourth, the lack of detailed information on polypharmacy, treatment intensity, or medication adherence constrains interpretation of multimorbidity management. Finally, although the survey is nationally representative, findings may not fully generalize to institutionalized populations, individuals with severe disabilities, or minority ethnic groups. Future longitudinal studies with objective clinical measurements are warranted to validate and extend these findings.

## Implications and recommendations

This study provides nationally representative evidence on the prevalence, patterns, and determinants of comorbidities among Iranian adults with diabetes, thereby contributing to the expanding knowledge base on multimorbidity. By identifying both the burden and distribution of comorbid conditions, the findings highlight the complexity of health challenges faced by this population and underscore the need to incorporate multimorbidity considerations into diabetes research and surveillance.

Because of its cross-sectional design, the study does not allow causal interpretation or direct translation into clinical or policy recommendations. Nevertheless, the descriptive patterns identified here can guide the development of future longitudinal and interventional studies aimed at clarifying disease trajectories, identifying modifiable risk factors, and assessing strategies to improve diabetes management in the context of multiple chronic conditions. Prospective studies with objective health assessments, detailed treatment data, and evaluation of targeted interventions will be essential for transforming these findings into practical approaches that enhance outcomes for individuals living with diabetes and multimorbidity.

## Conclusion

This study provides the first nationally representative assessment of comorbidity burden and patterns among Iranian adults with diabetes. Approximately 70% of diabetic patients in Iran had at least one comorbidity, and a substantial proportion had two or more. Older age, male sex, obesity, low education, and physical inactivity were key determinants of multimorbidity. Higher comorbidity burden was associated with poorer glycemic control, even after adjusting for key demographic and lifestyle factors.

These findings underscore the importance of adopting a comprehensive, integrated approach to diabetes care that considers the cumulative burden of chronic diseases rather than focusing solely on individual conditions. From a healthcare delivery perspective, these results support the implementation of integrated care models within primary care settings, with routine screening for common comorbidities among high-risk groups. Strengthening preventive care, improving early detection, and addressing modifiable risk factors are essential steps for mitigating the clinical and economic burden of diabetes-related multimorbidity in Iran.

## Supporting information

**S1 Fig. The proportion of missing values for each variable utilized in the study.**
(DOCX)

**S1 Table. Stepwise Multinomial Logistic Regression Analysis of Factors Associated with Comorbidity in the Female Diabetic Population.**
(DOCX)

**S2 Table. Stepwise Multinomial Logistic Regression Analysis of Factors Associated with Comorbidity in the Male Diabetic Population.**
(DOCX)

## Acknowledgments

We extend our sincere appreciation to our colleagues at the Non-Communicable Diseases Research Center (NCDRC), the EMRI, and the National Institute of Health Research of the Islamic Republic of Iran at Tehran University of Medical Sciences for their invaluable support and collaboration throughout this study. We also acknowledge the assistance of OpenAI's ChatGPT in providing language editing support, which enhanced the clarity and quality of the manuscript.

## Author contributions

**Conceptualization:** Samaneh Akbarpour, Samaneh Asgari, Nazila Rezaei.

**Formal analysis:** Sarmad Salehi.

**Methodology:** Samaneh Asgari.

**Project administration:** Nazila Rezaei.

**Supervision:** Nazila Rezaei.

**Validation:** Sina Azadnajafabad, Samaneh Akbarpour, Nazila Rezaei.

**Writing – original draft:** Koushan Hajiuni, Zahra Rajabli.

**Writing – review & editing:** Sarmad Salehi, Sina Azadnajafabad, Samaneh Akbarpour, Samaneh Asgari, Nazila Rezaei.

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
