## [Decision Letter · Decision Letter 0]

15 Aug 2025

Dear Dr. Rezaei,

Thank you for submitting your manuscript to PLOS ONE. After careful consideration, we feel that it has merit but does not fully meet PLOS ONE’s publication criteria as it currently stands. Therefore, we invite you to submit a revised version of the manuscript that addresses the points raised during the review process.

We look forward to receiving your revised manuscript.

Kind regards,

Patricia Khashayar

Academic Editor

PLOS ONE

Journal Requirements:

2. In the online submission form, you indicated that [The datasets generated and/or analyzed during the current study are not publicly available due to contractual agreements with the National Institute of Health Research, I.R. Iran. However, data may be made available from the corresponding author upon reasonable request and with permission from the funding body.].

Reviewers' comments:

Reviewer's Responses to Questions

**Comments to the Author**

1. Is the manuscript technically sound, and do the data support the conclusions?

Reviewer #1: Partly

Reviewer #2: Yes

2. Has the statistical analysis been performed appropriately and rigorously?

Reviewer #1: No

Reviewer #2: Yes

3. Have the authors made all data underlying the findings in their manuscript fully available?

Reviewer #1: Yes

Reviewer #2: Yes

4. Is the manuscript presented in an intelligible fashion and written in standard English?

Reviewer #1: Yes

Reviewer #2: No

Reviewer #1: Methodological Transparency

The study design needs clearer exposition. Specify inclusion/exclusion criteria, data sources, and whether it was retrospective or prospective.

Clarify the process of participant recruitment and sampling strategy.

Statistical Rigor

Clarify how potential confounders were handled. If multivariate models were used, specify which variables were adjusted for.

Include confidence intervals for key findings.

If subgroup analyses or multiple comparisons were conducted, please report how type I error risk was mitigated.

Discussion Section

Avoid overgeneralizing the findings. For instance, claims regarding policy implications should be made cautiously given the limitations.

Expand the limitations section to include issues such as potential bias, residual confounding, or limited generalizability.

Language and Style

While generally readable, the manuscript would benefit from professional English editing.

Avoid overly long or complex sentences, especially in the Results and Discussion.

Figures and Tables

Please ensure all figures are self-explanatory and include complete legends.

Abbreviations in tables should be defined in footnotes.

Reviewer #2: I reviewed the manuscript entitled “Prevalence and Factors Associated with Comorbidities in Iranian Patients with Diabetes: Insights from the 2021 STEPS Survey”. It is an interesting topic. The manuscript is in the scope of the journal, but certain shortcomings should be addressed before the article can be published:

• The manuscript should be edited for language, as it contains several grammatical mistakes.

• In the abstract, Add a brief info on inclusion and exclusion.

• Several abbreviations such as line 80 (T2D) and STEPS each abbreviation should be explained first then be used in the article.

• The results section needs more explanation.

• As it is requesting to publish in an international multidisciplinary Journal, in the conclusion you should include how the findings of this study can affect the general practice except for Iran ethnic groups population and some insight on future studies. The generalizability of the results should also be discussed.

**Do you want your identity to be public for this peer review?** For information about this choice, including consent withdrawal, please see our Privacy Policy

Reviewer #1: No

Reviewer #2: **Yes:** Pouria Khashayar

---

## [Author Response · Author response to Decision Letter 1]

27 Aug 2025

Thank you so much for your attention and your reviewers' opinions. The manuscript was checked and changes were performed. In addition, the responses to each of the comments are included in this file.

Kind Regards

Dr. Nazila Rezaei,

Corresponding author

Reviewer #1: Methodological Transparency

The study design needs clearer exposition. Specify inclusion/exclusion criteria, data sources, and whether it was retrospective or prospective.

Clarify the process of participant recruitment and sampling strategy.

Response: We sincerely thank the reviewer for this thoughtful and constructive comment. We highly appreciate the careful attention paid to the methodological clarity of our manuscript. In response, we have provided a more detailed exposition of the study design. Specifically, the inclusion and exclusion criteria, data sources, and the nature of the study (cross-sectional) have been explicitly described in the Study Design section. Furthermore, details regarding participant recruitment and the sampling strategy have been clarified and expanded in the Survey Sampling and Population section (see Page 5, lines 98-105 & Page 6, lines 115-126).

Statistical Rigor

Clarify how potential confounders were handled. If multivariate models were used, specify which variables were adjusted for.

Include confidence intervals for key findings.

If subgroup analyses or multiple comparisons were conducted, please report how type I error risk was mitigated.

Response: We thank the reviewer for this insightful comment, which helped us to clarify the analytical approach. In response, the Statistical Analyses section has been revised and expanded. We now explicitly describe how potential confounders were addressed by including relevant sociodemographic and health-related variables in the multivariable model. The variables considered for inclusion, the selection method, and the criteria for retention in the final models have been specified. In addition, we have clarified that both descriptive and regression analyses report 95% confidence intervals for the key findings to ensure the precision of estimates. Finally, we explicitly state that no formal subgroup analyses or multiple comparison adjustments were conducted; however, the primary analyses were predefined and hypothesis-driven, thereby minimizing the risk of type I error (see Page 8, lines 164-176).

Discussion Section

Avoid overgeneralizing the findings. For instance, claims regarding policy implications should be made cautiously given the limitations.

Expand the limitations section to include issues such as potential bias, residual confounding, or limited generalizability.

Response: We sincerely thank the reviewer for this thoughtful comment. In response, we added a more detailed discussion of study limitations, including the potential for recall and social desirability bias due to self-reported data, residual confounding from unmeasured variables, and restrictions in the generalizability of the findings to certain subpopulations. We revised the Implications and Recommendations section to avoid overgeneralization. Instead of proposing direct interventions or policy changes, we emphasized that our study provides descriptive evidence that contributes to a broader understanding of multimorbidity in diabetes and should be considered as one piece of the larger puzzle. The revised section highlights that our findings serve as a foundation for future longitudinal and interventional research rather than offering prescriptive recommendations. We believe these revisions address the reviewer’s concerns by clarifying the scope of our conclusions and presenting the limitations more comprehensively (see Page 22, lines 376-388).

Language and Style

While generally readable, the manuscript would benefit from professional English editing.

Avoid overly long or complex sentences, especially in the Results and Discussion.

Response: We appreciate the reviewer’s valuable comment regarding the language and readability of the manuscript. In response, the entire text has been carefully revised by a professional editor to improve sentence structure, grammar, and punctuation. Overly long or complex sentences, particularly in the Results and Discussion sections, were simplified to enhance clarity and readability.

Figures and Tables

Please ensure all figures are self-explanatory and include complete legends.

Abbreviations in tables should be defined in footnotes.

Response: We thank the reviewer for this valuable comment. In accordance with the suggestion, Figure 1 has been revised and supplemented with additional details to make the study population selection process clearer. The legend of Figure 2 has also been expanded to improve clarity. Furthermore, all abbreviations in the tables were carefully checked to ensure that they are fully defined in the corresponding footnotes.

Reviewer #2: I reviewed the manuscript entitled “Prevalence and Factors Associated with Comorbidities in Iranian Patients with Diabetes: Insights from the 2021 STEPS Survey”. It is an interesting topic. The manuscript is in the scope of the journal, but certain shortcomings should be addressed before the article can be published:

The manuscript should be edited for language, as it contains several grammatical mistakes.

Response: We appreciate the reviewer’s comment. The manuscript has been thoroughly revised by a professional editor to correct grammatical mistakes and improve overall language quality.

In the abstract, Add a brief info on inclusion and exclusion.

Response: We thank the reviewer for this helpful suggestion. In line with the comment, we have revised the abstract to briefly describe the inclusion and exclusion criteria (see Page 2, lines 31-32).

Several abbreviations such as line 80 (T2D) and STEPS each abbreviation should be explained first then be used in the article.

Response: Thank you for your comment. All abbreviations including T2D, STEPS, BMI, and HbA1c have now been defined at their first occurrence in the manuscript to ensure clarity and consistency.

The results section needs more explanation.

Response: We appreciate the reviewer’s valuable comment. In response, the entire Results section has been carefully reviewed and revised to provide clearer explanations and more detailed descriptions of the findings.

As it is requesting to publish in an international multidisciplinary Journal, in the conclusion you should include how the findings of this study can affect the general practice except for Iran ethnic groups population and some insight on future studies. The generalizability of the results should also be discussed.

Response: We sincerely thank the reviewer for this insightful comment. In response, we have revised the Conclusion section to emphasize the generalizability of our findings. Although the study was conducted in an Iranian population, the results provide implications that can be applicable to broader and international contexts. We also included remarks on how the findings may inform general practice beyond the Iranian ethnic groups and highlighted directions for future research (Page 21 lines 355-358 in discussion section & Page 23 lines 399-403 in Conclusion section).

Journal Requirements:

Response: We thank the reviewer for this important note. We have carefully reviewed the PLOS ONE manuscript preparation guidelines, including style and file naming requirements, and have revised our submission accordingly to ensure full compliance with the journal’s instructions.

2. In the online submission form, you indicated that [The datasets generated and/or analyzed during the current study are not publicly available due to contractual agreements with the National Institute of Health Research, I.R. Iran. However, data may be made available from the corresponding author upon reasonable request and with permission from the funding body.].

Response: We thank the editor for raising this important point regarding data availability. In accordance with the journal’s policy, we have revised the Data Availability Statement in the manuscript. The dataset used in this study is not publicly available, as it is the property of the National Institute for Health Research (NIHR) at Tehran University of Medical Sciences, and access is restricted by institutional policies to ensure the privacy of participants. However, data may be made available from the corresponding author upon reasonable request and subject to approval by NIHR (contact: nihr@tums.ac.ir).

Response: We appreciate the reviewer’s guidance. The ethics statement has been removed from all other sections and is now reported exclusively in the Methods section of the manuscript.

---

## [Decision Letter · Decision Letter 1]

9 Nov 2025

Dear Dr. Rezaei,

We look forward to receiving your revised manuscript.

Kind regards,

Patricia Khashayar

Academic Editor

PLOS ONE

Journal Requirements:

Reviewers' comments:

Reviewer's Responses to Questions

**Comments to the Author**

Reviewer #3: (No Response)

2. Is the manuscript technically sound, and do the data support the conclusions?

Reviewer #3: Partly

3. Has the statistical analysis been performed appropriately and rigorously?

Reviewer #3: No

4. Have the authors made all data underlying the findings in their manuscript fully available?

Reviewer #3: Yes

5. Is the manuscript presented in an intelligible fashion and written in standard English?

Reviewer #3: Yes

Reviewer #3: This study analyzes nationally representative data from the 2021 Iran STEPS survey to assess the prevalence and determinants of comorbidities among adults with diabetes. The topic is timely and of clear public-health relevance, particularly given the growing burden of diabetes and associated conditions in low- and middle-income settings. The manuscript offers useful descriptive evidence and fills an important regional data gap; however, several methodological and interpretive issues.

Comments:

1. Methods: Citation [11] appears to refer to a study that utilized the Iran STEPS survey rather than a primary description of the survey methodology itself. It would strengthen the manuscript if the authors could include an appendix briefly outlining how the STEPS survey was designed and implemented. For example, the U.S. MEPS data provide a dedicated summary of the Panel Design and Data Collection Process (https://meps.ahrq.gov/mepsweb/survey_comp/hc_data_collection.jsp), , which helps readers understand how the survey was conducted. The authors are also encouraged to report the overall response rate of the 2021 Iran STEPS survey, in addition to the number of individuals ultimately included in the analysis. The authors mention survey weights were “adjusted for non-response within age groups” and “poststratification weighting by age, sex, and residence”, but they do not specify the base weights, how they were derived from the cluster sampling, or how province-level sampling probabilities were handled. It’s unclear if they accounted for stratification and clustering in variance estimation. These should be provided in appendix.

2. They rely partly on self-reported diagnoses for CVD and cancer but laboratory or clinical cutoffs for diabetes, HTN, and CKD. This inconsistency could lead to differential misclassification bias across conditions. The authors should discuss this in the Discussion.

3. The authors note “complete-case analysis”. However, they should also describe the proportion or pattern of missingness for each variables, whether data were missing at random or not, or if any sensitivity analyses were performed. A table summarizing missing data by variable in appendix, and justification for using complete-case analysis are recommended.

4. Why a multinomial rather than an ordered logistic model was chosen (the number of comorbidities is ordinal: 0, 1, ≥2)? Additionally, it is important to discuss if there is multicollinearity among predictors (e.g., BMI, waist circumference, physical activity, education)? If so, would it affect the author’s findings and conclusions?

5. The authors seem omit several plausible covariates available in Iran STEPS data. For example, they mention a “diet score” descriptively but not in the model. Did the data and the models include medication use beyond antihypertensives or glucose-lowering drugs?

**Do you want your identity to be public for this peer review?** For information about this choice, including consent withdrawal, please see our Privacy Policy

Reviewer #3: No

---

## [Author Response · Author response to Decision Letter 2]

29 Nov 2025

Reviewer #3: This study analyzes nationally representative data from the 2021 Iran STEPS survey to assess the prevalence and determinants of comorbidities among adults with diabetes. The topic is timely and of clear public-health relevance, particularly given the growing burden of diabetes and associated conditions in low- and middle-income settings. The manuscript offers useful descriptive evidence and fills an important regional data gap; however, several methodological and interpretive issues.

Comments:

1. Methods: Citation [11] appears to refer to a study that utilized the Iran STEPS survey rather than a primary description of the survey methodology itself. It would strengthen the manuscript if the authors could include an appendix briefly outlining how the STEPS survey was designed and implemented. For example, the U.S. MEPS data provide a dedicated summary of the Panel Design and Data Collection Process (https://meps.ahrq.gov/mepsweb/survey_comp/hc_data_collection.jsp), which helps readers understand how the survey was conducted. The authors are also encouraged to report the overall response rate of the 2021 Iran STEPS survey, in addition to the number of individuals ultimately included in the analysis. The authors mention survey weights were “adjusted for non-response within age groups” and “poststratification weighting by age, sex, and residence”, but they do not specify the base weights, how they were derived from the cluster sampling, or how province-level sampling probabilities were handled. It’s unclear if they accounted for stratification and clustering in variance estimation. These should be provided in appendix.

Response: We appreciate the reviewer’s careful attention to the methodological details. We have revised the manuscript to incorporate the requested clarifications and enhance transparency. We would like to clarify that Reference 11 in our manuscript (in the revised manuscript reference 10) is the official study protocol of the 2021 Iran STEPS survey, which provides the primary description of the survey design. This protocol includes detailed information on the sampling strategy, target population, fieldwork procedures, and data-collection processes (DOI: 10.34172/aim.2022.99). Furthermore, the supplementary material accompanying the protocol provides a comprehensive explanation of the weighting methodology, including computation of base weights, non-response adjustments, and post-stratification procedures. The supplementary file is publicly available at:

https://pmc.ncbi.nlm.nih.gov/articles/instance/10685773/bin/aim-25-634-s002.pdf. As outlined in this supplementary document, the weighting strategy explicitly accounts for the design effect, and variance estimation incorporates clustering, consistent with the complex multistage sampling framework of STEPS 2021. Because these methodological details have already been thoroughly documented in the official protocol and its supplementary material, we did not repeat them in our manuscript or its appendix. However, to enhance clarity and address your specific points, we have made the following additions to the manuscript:

1. In the Methods section (Page 6, Lines 115-116), we have now explicitly reported that the overall response rate of the 2021 Iran STEPS survey was 97.73%.

2. In the Statistical Analysis section (Page 9, lines 199-200), we have added a sentence explicitly stating that design effect and cluster-based variance estimation were accounted for in all analyses.

2. They rely partly on self-reported diagnoses for CVD and cancer but laboratory or clinical cutoffs for diabetes, HTN, and CKD. This inconsistency could lead to differential misclassification bias across conditions. The authors should discuss this in the Discussion.

Response: We thank the reviewer for highlighting this important point regarding potential misclassification. To address this, we have added a sentence in the Discussion to clearly acknowledge this limitation. The added sentence reads: "It should also be noted that CVD and cancer were based on self-reported diagnoses, whereas HTN, CKD, and diabetes were identified through clinical or laboratory criteria. This difference in disease ascertainment may potentially lead to an underestimation of CVD and cancer and influencing the relative distribution of comorbidity types." (Page 18, Lines 315-319). This addition clarifies the potential impact of differing disease ascertainment methods on the reported comorbidity patterns and ensures transparency regarding this methodological limitation.

3. The authors note “complete-case analysis”. However, they should also describe the proportion or pattern of missingness for each variables, whether data were missing at random or not, or if any sensitivity analyses were performed. A table summarizing missing data by variable in appendix, and justification for using complete-case analysis are recommended.

Response: Thanks for the reviewer’s comment. Given that the missing data for covariates was <10% (maximum 5.1%) as shown in Fig S1, complete case analysis was employed, as the imputation may offer little advantage [1, 2] (Page 7, Lines 137-139). However, as a sensitivity analysis multiple imputation with 10 iteration was done and results remained unchanged (data not shown).

4. Why a multinomial rather than an ordered logistic model was chosen (the number of comorbidities is ordinal: 0, 1, ≥2)?

Response: We appreciate the reviewer's valuable question regarding the choice of our regression model. The choice of using multinomial stepwise logistic regression over ordinal logistic regression depends to the nature of the dependent variable. Ordinal logistic regression is appropriate when the dependent variable is ordinal by nature (e.g., categories that have a natural order). If the categories (0, 1, more than 2) were treated as nominal (without a natural order), multinomial logistic regression would be more suitable. In our cases there is no order between prevalence of HTN, CVD, CVD, or cancer. Moreover, Ordinal logistic regression assumes that the relationship between each pair of outcome groups is the same (the proportional odds assumption). While it’s not true in our data, therefore multinomial logistic regression may be preferred [3, 4].

Additionally, it is important to discuss if there is multicollinearity among predictors (e.g., BMI, waist circumference, physical activity, education)? If so, would it affect the author’s findings and conclusions?

Response: We have assessed multicollinearity among predictors in our models. All variance inflation factors (VIFs) were below 2, indicating no significant multicollinearity. You can find each variable’s VIF in the table below:

Variable Categories VIF

Age 1.36

Sex 1.78

Residential area 1.19

Years of schooling

1_6 vs. illiterate 1.71

7_11 vs. illiterate 1.63

≥12 vs. illiterate 1.98

BMI

25-30 vs. <25 1.90

>30 vs. <25 1.97

Marital status 1.18

Employment status 1.58

Physical activity 1.06

Wealth index

2nd quintile vs. 1st quintile 1.61

3rd quintile vs. 1st quintile 1.63

4th quintile vs. 1st quintile 1.79

5th quintile vs. 1st quintile 1.96

Smoking 1.10

Ever alcohol consumption 1.05

Basic health insurance 1.02

Glucometer at home 1.09

Diet score

2nd tertile vs. 1st tertile 1.39

3rd tertile vs. 1st tertile 1.54

Given the logical and empirically observed collinearity between BMI and waist circumference, only BMI was included in the final model. Additionally, we have added a sentence in the Methods section to explicitly state that multicollinearity was checked using VIFs (Page 10, Lines 216-218).

5. The authors seem omit several plausible covariates available in Iran STEPS data. For example, they mention a “diet score” descriptively but not in the model. Did the data and the models include medication use beyond antihypertensives or glucose-lowering drugs?

Response: We appreciate the reviewer’s insightful comment. In this study, all variables available in the national STEPS survey that could plausibly be associated with our outcome among individuals with diabetes were considered. The diet score was also included as one of the candidate variables; as shown in Table 1, its distribution was examined descriptively. This variable was initially entered into the multinomial logistic regression model; however, it did not remain in the final multivariable model because it was not statistically significant in the stepwise selection procedure. Regarding medication use, only glucose-lowering medications and antihypertensive drugs were included, because other medications captured in STEPS were not relevant to the diseases under investigation in this study.

References:

[1] P. Cummings, "Missing data and multiple imputation," (in eng), JAMA pediatrics, vol. 167, no. 7, pp. 656-61, Jul 2013, doi: 10.1001/jamapediatrics.2013.1329.

[2] F. Barzi and M. Woodward, "Imputations of missing values in practice: results from imputations of serum cholesterol in 28 cohort studies," (in eng), American journal of epidemiology, vol. 160, no. 1, pp. 34-45, Jul 1 2004, doi: 10.1093/aje/kwh175.

[3] C. Kwak and A. Clayton-Matthews, "Multinomial logistic regression," Nursing research, vol. 51, no. 6, pp. 404-410, 2002.

[4] P. Warner, "Ordinal logistic regression," Journal of Family Planning and Reproductive Health Care, vol. 34, no. 3, p. 169, 2008.

---

## [Decision Letter · Decision Letter 2]

2 Feb 2026

Dear Dr. Rezaei,

Thank you for submitting your manuscript to PLOS ONE. After careful consideration, we feel that it has merit but does not fully meet PLOS ONE’s publication criteria as it currently stands. Therefore, we invite you to submit a revised version of the manuscript that addresses the points raised during the review process.

We look forward to receiving your revised manuscript.

Kind regards,

Patricia Khashayar

Academic Editor

PLOS One

Journal Requirements:

Reviewers' comments:

Reviewer's Responses to Questions

**Comments to the Author**

Reviewer #2: All comments have been addressed

2. Is the manuscript technically sound, and do the data support the conclusions?

Reviewer #2: Yes

3. Has the statistical analysis been performed appropriately and rigorously?

Reviewer #2: Yes

4. Have the authors made all data underlying the findings in their manuscript fully available?

Reviewer #2: Yes

5. Is the manuscript presented in an intelligible fashion and written in standard English?

Reviewer #2: No

Reviewer #2: (No Response)

**Do you want your identity to be public for this peer review?** For information about this choice, including consent withdrawal, please see our Privacy Policy

Reviewer #2: No

---

## [Author Response · Author response to Decision Letter 3]

7 Feb 2026

We would like to sincerely thank the reviewers for their careful reading of our manuscript and for their insightful and constructive comments. We have revised the manuscript accordingly, and our point-by-point responses are provided below.

Kind Regards

Dr. Nazila Rezaei,

Corresponding author

Title: Prevalence and Factors Associated with Comorbidities in Iranian Patients with Diabetes: a national study

Submission ID: PONE-D-25-25729R1

Review

I reviewed the manuscript “Prevalence and Factors Associated with Comorbidities in Iranian Patients with type 2 Diabetes: a national study”. This manuscript addresses an important and timely issue—the prevalence and determinants of comorbidities among Iranian patients with Type 2 Diabetes Mellitus (T2DM). The study's focus on comorbidities such as hypertension, CKD, CVD, and cancer, and their association with sociodemographic factors, offers comprehensive insights into the burden of multimorbidity in this population.

The manuscript is within the scope of the journal, but certain main shortcomings should be addressed before as below:

• The manuscript should be revised for language accuracy, as it contains several grammatical errors.

Response: We thank the reviewer for highlighting the need to improve language accuracy in the manuscript. In response, we have thoroughly reviewed the entire article from the beginning and corrected all grammatical errors that were identified.

• Currently, the manuscript lacks detailed prevalence estimates for individual comorbidities and their clustering patterns, which are crucial for comprehending the overall burden.

Response: Thank you for the valuable comment. We have added the overall prevalence of each comorbidity in the entire cohort (HTN: 63.46%, CKD: 27.44%, CVD: 20.61%, Cancer: 1.88%). This, together with the clustering patterns already presented, now provides a complete picture of the comorbidity burden. The results section has been updated accordingly (Page 11, lines 232-236).

• Although the multinomial logistic regression approach is suitable, investigating interaction effects (e.g., between age and BMI or socioeconomic status and physical activity) could yield additional valuable insights.

Response: Thank you for your insightful comment and constructive feedback. As requested, we have examined the potential interactions between age and sex with the other covariates in our model. Since there were no significant interaction between sex, age and other covariates, all analysis was done in the whole population (Page 10-11, lines 222-226).

• Conducting sensitivity analyses, such as stratification by age groups or adjusting for the duration of diabetes, might improve the depth of understanding.

Response: We thank the reviewer for the suggestion. In our primary analysis, we examined interactions between age, sex and other variables, none of which were statistically significant. Due to limited statistical power, a stratified analysis by age was not performed, and the results are presented in aggregate. Furthermore, to enhance the study results, we have included sex-stratified results in the supplementary material (Table S1 & Table S2). Information on the duration of diabetes was not available in the dataset, which is noted in the Limitations section (Page 23, line 420).

• Consistent reporting of percentages and confidence intervals, along with correction of minor typographical errors, would enhance the manuscript's professionalism.

Response: We thank the reviewer for pointing out the need for consistent reporting. We have ensured uniform formatting of percentages and confidence intervals throughout the manuscript and corrected all minor typographical errors.

• The discussion should consider potential unmeasured confounders, such as medication adherence, healthcare access, or genetic factors, which could influence patterns of comorbidity and glycemic control.

Response: Thank you for this insightful comment. We have revised the Discussion section to explicitly acknowledge the potential impact of unmeasured confounders, including medication adherence, access to healthcare services, treatment intensity, and genetic predisposition, on comorbidity patterns and glycaemic control. We also noted that the lack of these variables may result in residual confounding and should be considered when interpreting the findings (Page 21, lines 345-350).

• It could also expand on how these findings should shape healthcare delivery, emphasizing integrated care models, routine screening in primary care settings, and health education initiatives tailored to high-risk populations.

Response: Thank you for this valuable suggestion. We have revised the Conclusion to explicitly highlight the implications of our findings for healthcare delivery, emphasizing the role of integrated care models within primary care settings and routine screening for comorbidities among high-risk populations (Page 26, lines 458-460).

---

## [Editor Report · Decision Letter 3]

11 Feb 2026

Prevalence and Factors Associated with Comorbidities in Iranian Patients with type 2 Diabetes: a national study

PONE-D-25-25729R3

Dear Dr. Rezaei,

We’re pleased to inform you that your manuscript has been judged scientifically suitable for publication and will be formally accepted for publication once it meets all outstanding technical requirements.

Kind regards,

Patricia Khashayar

Academic Editor

PLOS One
---

## [Editor Report · Acceptance letter]

PONE-D-25-25729R3

PLOS One

Dear Dr. Rezaei,

I'm pleased to inform you that your manuscript has been deemed suitable for publication in PLOS One. Congratulations! Your manuscript is now being handed over to our production team.

Kind regards,

on behalf of

Dr. Patricia Khashayar

Academic Editor

PLOS One